# Prognostic Role of Clinical Features of Moderate Forms of COVID-19 Requiring Hospitalization

**DOI:** 10.3390/jpm13060900

**Published:** 2023-05-26

**Authors:** Antigona Carmen Trofor, Andrei Tudor Cernomaz, Lucia Maria Lotrean, Radu Adrian Crișan-Dabija, Jose L. Penalvo, Oana Elena Melinte, Daniela Robu Popa, Milena Adina Man

**Affiliations:** 1Discipline of Pneumology, III-rd Medical Department, Faculty of Medicine, “Grigore T. Popa” University of Medicine and Pharmacy, 700115 Iași, Romania; antigona.trofor@umfiasi.ro (A.C.T.);; 2Discipline of Hygiene, Department of Community Medicine, “Iuliu Hațieganu” University of Medicine and Pharmacy, 400012 Cluj, Romania; 3Unit of Non-Communicable Diseases, Department of Public Health, Institute of Tropical Medicine, 2000 Antwerp, Belgium; 4Discipline of Pneumology, Department of Medical Sciences, Faculty of Medicine, “Iuliu Hațieganu” University of Medicine and Pharmacy, 400012 Cluj, Romania

**Keywords:** COVID-19, clinical features, prognostic

## Abstract

Introduction: We aimed to characterize the clinical features of moderate forms of COVID-19 requiring hospitalization and potentially identify predictors for unfavorable outcomes. Methods: Pooled anonymized clinical data from 452 COVID-19 patients hospitalized in two regional Romanian respiratory disease centers during the Alpha and Delta variant outbreaks were included in the analysis. Results: Cough and shortness of breath were the most common clinical features; older patients exhibited more fatigue and dyspnea and fewer upper airway-related symptoms such as smell loss or sore throat. The presence of confusion, shortness of breath and age over 60 years were significantly associated with worse outcomes (odds ratios 5.73, 2.08 and 3.29, respectively). Conclusion: The clinical picture on admission may have a prognostic role for moderate forms of COVID-19. Clear clinical definitions and developing adequate informational infrastructure allowing complex data sharing and analysis might be useful for fast research response should a similar outbreak occur in the future.

## 1. Introduction

The SARS-CoV-2 outbreak was officially declared by the World Health Organization in March, 2020, and the medical and public health response were hampered by a relative lack of knowledge. Clinical manifestations of SARS-CoV-2 infection proved heterogeneous, multisystemic and of variable magnitude, thus complicating the task of identifying potential cases and assessing associated risks. Guidelines and risk assessment tools have been developed and implemented, but clinical features remain the first resource used in medical practice to drive other subsequent interventions. The COVID-19 medical picture is still evolving as new mechanisms and medical implications are brought to light.

We aimed to characterize the clinical picture of moderate forms of SARS-CoV-2 requiring advanced medical care and to identify the potential prognostic value of various clinical features.

## 2. Materials and Methods

We analyzed data from SARS-CoV-2 patients originating from the 2 Romanian sites contributing to the unCover network [1]—‘Unravelling Data for Rapid Evidence-Based Response to COVID-19’, an international research grant developed under the umbrella of the H2020-SC1-PHE-CORONAVIRUS-2020 call set at an early stage of the COVID-19 pandemic in 2020. This initiative created a research body network (29 members from 18 countries) aimed at collecting, sharing, and analyzing COVID-19 data. The infrastructure was designed to enable research initiatives on risk factors, safety, treatments, outcomes and potential strategies against COVID-19 in real-world settings by facilitating access to otherwise scattered and differently structured datasets and by developing computational and analytical platforms to streamline pooled data analysis.

Original medical data were extracted from electronic and physical medical records of patients hospitalized for moderate forms of SARS-CoV-2 infection between May 2020 and October 2021, corresponding to the Alpha and Delta variant outbreaks, and shared within the unCover network using a harmonized dataset structure.

The parameters included in the analysis were gender, age, clinical features on admission, known underlying medical conditions, length of hospital stay and outcome.

Clinical features were represented as dichotomous variables reflecting the presence of cough, confusion, anorexia, abdominal pain, arthralgia, chest pain, diarrhea, fatigue, fever, headache, myalgia, nausea, nasal discharge, shortness of breath, convulsions, smell loss, sore throat, skin rash, taste loss and wheezing.

Limited co-morbidity data were available as binary variables encoding the presence of arterial hypertension, cardiovascular disease (other than hypertension), chronic obstructive pulmonary disease (COPD), asthma, chronic renal disease, chronic hepatic disease, chronic neurological conditions, dementia, rheumatological disorders and malignant disease.

The outcome was defined as favorable (patient recovered and discharged from hospital) or unfavorable (death occurred during hospitalization or patient was transferred to other services due to complications).

Data sharing, manipulation and analysis was conducted in accordance with relevant local and European ethics guidelines following the methodology outlined in the unCover protocols (1) and maintaining complete anonymity of the original datasets. Data were analyzed using the tools built into the unCover platform and SPSS v 10 for a consolidated local database.

## 3. Results

In total, 452 anonymized medical records originating from patients hospitalized with moderate SARS-CoV-2 disease between May 2020 and October 2021 were included in the analysis; 246 cases (54.4%) were males. The mean age was 61.8 ± 13.9 years and the range was 20–92 years.

The mean hospital stay was 12.7 ± 7.4 days; the majority—360 cases (79.6%)—recovered, 47 cases (10.4%) were transferred to other services (with medical complications) and 45 (10%) died.

The clinical features on admission are presented in Table 1—the most common was cough.

There were no statistically significant differences between gender subgroups. There were significant differences in terms of outcome only for shortness of breath at admission and confusion between stratified subgroups (chi square)—see Table 2.

The dataset was age-stratified using a threshold value of 60 years, and there were significant differences between subgroups regarding clinical features at admission. Shortness of breath was significantly more prevalent among elderly patients and sore throat among younger cases—see Table 3. Differences regarding fatigue, fever, smell and taste loss were considered as possibly significant (not significant after Bonferroni correction).

Various pre-existing morbid conditions were identified within the dataset pool—only 40 records (8.8%) showed no underlying conditions. The median value was two pre-existing disorders, and there were significant differences between age groups (*p* < 0.001, chi square)—see Figure 1.

The most prevalent underlying medical conditions in the general dataset were arterial hypertension and other cardiovascular disorders, followed by obesity and diabetes—see Figure 2. The prevalence of COPD, diabetes, hypertension, cardiovascular, renal, hepatic, neurological and rheumatological disease were significantly higher in the over 60 years of age group (chi square).

Considering the interim analysis results, a logistic regression model was used to characterize the relationship between the presence of two clinical features at admission (shortness of breath and confusion) and age group and the potential outcome: discharged vs. unfavorable (transfer with complications/death). A summary of the model is presented in Table 4.

Significant differences in pre-existing conditions were found when stratifying the data pool according to the presence of confusion and shortness of breath, as shown in Table 5.

## 4. Discussion

Fever and cough were repeatedly reported as the most frequently presenting symptoms for SARS-CoV-2 patients—a Pakistani retrospective study including 845 confirmed SARS-CoV-2 cases reported incidences of 72.7% and 59,5%, respectively [2]. These figures may vary in different reports: a Chinese study reported that more than half of patients had mild fever (less than 38 °C) [3], which may explain differences in fever incidence. Similarly, a retrospective study that included 7614 SARS-CoV-2 patients reported fever at admission in 50% of the cases, although 78.5% developed fever during the course of the disease [4]. Fever at admission was deemed to not have a prognostic role, although high fever during hospitalization was associated with increased mortality. The authors remarked that low body temperature at admission was a poor prognosis factor.

Our dataset shows cough to be the most frequent symptom (69%), and fever is in fourth place after shortness of breath and fatigue—these differences may be explained by the population included and population testing policies (symptom-driven in some studies). Variability in measurement technique, body site and instrumental bias [5] may also explain differences in fever-related figures; there are some published efforts to ensure a more consistent approach [6].

Shortness of breath was present at admission in more than half of the cases included in our analysis—unfortunately, there were no other details available, such as intensity.

The presence of shortness of breath at admission was associated with worse outcomes in our dataset; this was to be expected, as dyspnea was included in a COVID risk score alongside age, hemoptysis, cancer history, neutrophil to lymphocyte ratio, lactate dehydrogenase, chest radiography abnormality and unconsciousness [7]. A systematic review and meta-analysis that included 1934 mild and 1644 severe COVID cases from 19 published papers reported a dyspnea odds ratio of 3.28 (2.09–5.15) to developing a severe form. Fatigue, dizziness, and anorexia were also associated with worse outcomes [8]. Identifying the exact underlying mechanism of dyspnea is particularly difficult in clinical settings, as multiple phenomena may co-exist and interfere. Our data support an association between shortness of breath at admission and cardiovascular disfunction, particularly hypertension; the presence of obstructive respiratory disease seemed to be a less frequent occurrence.

Apart from their potential prognostic role, dyspnea and cough were reported as clinical features for 24% and 19% of patients with post-acute COVID symptoms [9].

The presence of confusion/disorientation was associated with worse prognosis in our dataset; a meta-analysis including data from 145,721 patients in 350 published studies found a prevalence of confusion/delirium of 34% in patients aged over 60 years and reported an association with increased mortality (odds ratio 1.80, 1.11–2.91) [10]. There are multiple mechanisms that may explain delirium in SARS-CoV-2 patients: hypoxia, systemic inflammatory response and also direct viral involvement—encephalitis, either primary or immune, has been reported [11] and is likely explained by the presence of the ACE 2 receptor on neurons and glial cells. In our data pool, confusion and disorientation were significantly associated with previous neurological disease, likely reflecting inadequate cerebral perfusion and pre-existing compensated conditions; the association with pre-existing hepatic conditions may suggest some form of toxic encephalopathy. The cerebral perfusion mechanism is supported by the possible significant association with pre-existing cardiovascular disorders. Similarly, existing prior psychiatric conditions were not significantly associated with the presence of confusion and disorientation in our study group.

SARS-CoV-2 infection is sometimes accompanied by headache; a meta-analysis including 14,275 patients from 86 independent studies reported a prevalence of 10.1% [12].

Headache prevalence among SARS-CoV-2 patients was the subject of a meta-analysis that included data from over 14,000 cases and was reported as 10.1%. No significant predictor role was identified, although the incidence of headache was slightly higher in severe or critical patients. Nevertheless, there are published data associating the presence of headache as an early symptom with a benign course of disease, younger age and lower levels of C-reactive protein [13]. Even so, the presence of headache at the onset of SARS-CoV-2 infection has been linked to fatigue and persistent headache as post-COVID complications [14] according to a study including 1100 patients that reported a headache prevalence of 18.6%. SARS-CoV-2 headache has been characterized as bilateral, long-lasting, resistant to analgesics and also potentially associated with dysgeusia/anosmia and gastrointestinal complaints [15].

SARS-CoV-2 headache seems particularly difficult to characterize, as there are other systematic reviews giving a large range for its incidence (10 to 70%) and reporting various at-risk groups (females and younger patients) [16]. Multiple etiologic mechanisms have been considered [17]: increased cytokine levels, viral involvement of trigeminal nerve, hypoxia and increased cerebrospinal fluid pressure. The immune hypothesis is supported by reports on post-vaccinal headache [18].

Smell loss and smell alteration are well-known clinical features of upper airway viral infections; SARS-CoV-2-associated smell loss typically presents with minimal discharge or congestion—an aspect underlined by possibly the first case report concerning this problem [19] and later confirmed by larger studies [20]. Our data confirm this pattern—11% presented with altered smell, while only 3.3% presented with notable nasal discharge; such clinical features support the hypothesis of a neurosensorial lesion—microvascular [21] and sustentacular cell [22] injury were reported as possible mechanisms. The real incidence of COVID-19 smell loss might be higher; some studies reported an incidence of around 40% [23] and a meta-analysis reported the incidence of smell disturbances in the range 4.9–85.6% [24].

Such ranges and variability might be partially explained by differences in clinical assessment—smell loss is likely considered of little significance in emergency services and possibly overlooked, although it can be objectively assessed. The real-life origin and retrospective nature of our dataset possibly explain the low incidence we found. There are reports ascribing a positive prognostic role for smell loss, possibly explained by a younger age predominance [25]. On the other hand, some cohort studies show no difference in terms of outcomes [26].

Musculoskeletal symptoms have been reported to occur in respiratory viral infections—our data suggest a higher incidence of myalgia (21%) than arthralgia (9.9%) for patients with moderate forms of COVID-19.

A single-center study reported the incidence of musculoskeletal symptoms to be 30% among 294 hospitalized patients with myalgia, arthralgia, and generalized body ache [27]—the distribution and number of affected sites was variable (mono-, oligo-, poliarthropathy, sometimes unilateral). Musculoskeletal involvement was generally not deemed to be associated with evolution to viral pneumonia.

There are data linking myalgia to an increased risk of ARDS [28]—at least as part of a predictive model together with elevated alanine aminotransferase and elevated hemoglobin, possibly reflecting systemic inflammation phenomena. However, a literature review disputes this association [29].

A large cross-sectional survey [30] including 3222 participants found myalgia to be more common than arthralgia, predominantly during the acute phase of the disease but sometimes persisting after clinical recovery, and reported some predictors such as hospitalization, sore throat, fatigue and ageusia.

A clear mechanism for COVID-19-associated arthralgia was not identified, but cases of post-COVID-19 reactive arthritis have been reported [31], and a possible autoimmune/molecular mimicry mechanism has been proposed [32]. More than one mechanism may be involved, as multiple patterns of involvement have been reported. It is important to note that musculoskeletal symptoms are the most prevalent features of post-COVID-19 syndrome—a systematic review reported that out of 100 patients, 66 had persistent myalgia and 52 persistent arthralgia [33].

Digestive symptoms such as diarrhea or abdominal pain were not associated with an adverse outcome in our dataset.

SARS-CoV-2-associated diarrhea incidence has been initially reported [34] in around 4% of cases, but there are published data suggesting higher values; a systematic review that included over 60,000 cases from 88 relevant studies estimated a pooled prevalence of 16.93% and found a potential prognostic role. The presence of diarrhea is associated with an odds ratio of 1.71 (1.31–2.24) for developing a severe form of the disease. The same systematic review found a prevalence of 8.89% for abdominal pain and 25.13% for loss of appetite.

The direct viral involvement of digestive tract cells is an accepted pathogenic mechanism as they express ACE2 receptors [35], and there are published data, albeit spurious, of clinical improvement after antiviral therapy [36]. Immune-mediated mucosal lesions and alterations to the gut microbiota have also been considered [37].

Chest pain usually has a cardiovascular, pleural or muscular origin. No data were available in our set, such as character, duration, irradiation or intensity, to further stratify and analyze.

There are data suggesting a prognostic role for chest pain—an odds ratio of 10.8 was reported in a 90-patient Chinese cohort for developing severe/critical SARS-CoV-2 pneumonia [38]. We found no prognostic role of chest pain in terms of outcome for moderate forms of SARS-CoV-2.

There are case reports of SARS-CoV-2 patients presenting with chest pain and high troponin levels without significant myocardial damage and with favorable outcome—possibly reflecting acute myocarditis [39].

Similarly, a German cohort of 100 recovered COVID-19 patients was investigated using cardiac magnetic resonance; the authors report cardiac involvement in 78% and ongoing myocardial inflammation in 60% [40]. Although these figures might overestimate the incidence of real viral myocarditis, they support the hypothesis of the cardiac nature of SARS-CoV-2 chest pain [41]. SARS-CoV-2 acute myocarditis incidence is estimated at 2.4 per 1000 hospitalizations among patients requiring hospitalization and typically occurs in the absence of pneumonia [42].

It was also reported that patients with pre-existing cardiac conditions have a higher risk of myocardial injury during the course of SARS-CoV-2 infections [43]. Taking all this into account, chest pain should always be carefully investigated in SARS-CoV-2 patients, although it may not always have a sinister implication.

No cutaneous manifestations were recorded in our dataset either, reflecting the late onset of such lesions or due to under reporting—a phenomenon known to occur in non-dermatological settings [44].

Age seems to play an important role as a prognostic factor, both regarding outcome and clinical picture of the SARS-CoV-2 infection. Age over 60 years seems to be associated with a higher incidence of cardiovascular clinical manifestations such as shortness of breath and fatigue; smell and taste disturbances and fever were rarer in the elderly subgroup.

Differences in clinical manifestations between elderly and younger patients may be explained by immune system age-related changes and by more sophisticated mechanisms such as altered expression of ACE2 receptors [45]. The association between age and respiratory distress is also supported by a Chinese study including 339 patients with a mean age of 71 ± 8 years with moderate, severe and critical forms of SARS-CoV-2 infections that found dyspnea, lymphocytopenia, chronic cardiovascular and respiratory conditions and the presence of respiratory distress syndrome to be predictors of poor outcome [46].

At least some differences between elderly and younger patients might be explained by underlying conditions; our data pool shows significant differences between the number and the spectrum of existing co-morbidities.

Considering this, age should be taken into account when defining outcomes—there are data recommending paying particular attention to non-lethal outcomes in younger populations as their magnitude may be underestimated [47].

Evident neurological impairment, age higher than 60 years and the presence of shortness of breath at admission emerge as risk factors for adverse outcomes (defined as death or medical complications) for the population under study—Delta variant SARS-CoV-2 patients requiring hospitalization. There are some limits that should be taken into account when extrapolating these results: the impact of co-morbid conditions was difficult to assess as only general limited data were available, the presented data refer to the Alpha and Delta outbreaks and may not reflect the characteristics of other viral strains and the term ‘requiring hospitalization’ is loosely defined, as no generally accepted criteria for admission were identified—some bias stemming from local policies and bed availability during the outbreak is to be expected.

## 5. Conclusions

The presence of shortness of breath and neurological impairment on hospital admission and age over 60 years seem to be associated with a worse prognosis in moderate forms of SARS-CoV-2 infections requiring hospitalization. Patients’ presentation of clinical features of COVID-19 seems to be age-related and partially explained by underlying co-morbid conditions.

Establishing a clear clinical picture of a novel disease was made more difficult by definition variability—many clinical features are not standardized, and observer and procedural bias is to be expected when using real-life data.

Creating and maintaining comprehensive clinical databases using clear definitions might facilitate research initiatives should a similar outbreak occur again. Database interoperability and harmonization allow data pooling and more robust analysis and should be encouraged; data privacy hurdles could be circumvented by various federated database implementations.

## Figures and Tables

**Figure 1 jpm-13-00900-f001:**
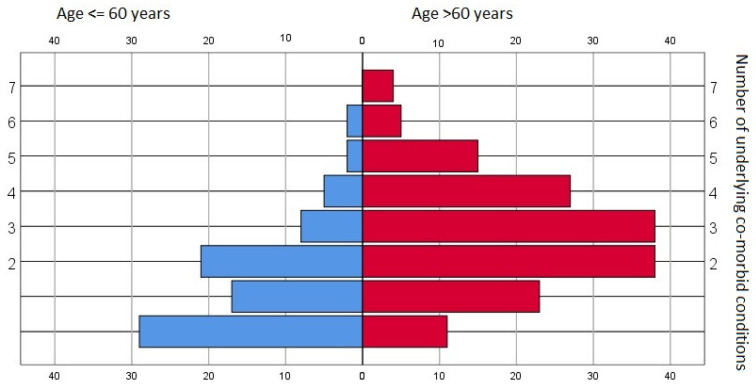
Prevalence of underlying co-morbid conditions—general data pool and age-stratified subgroups.

**Figure 2 jpm-13-00900-f002:**
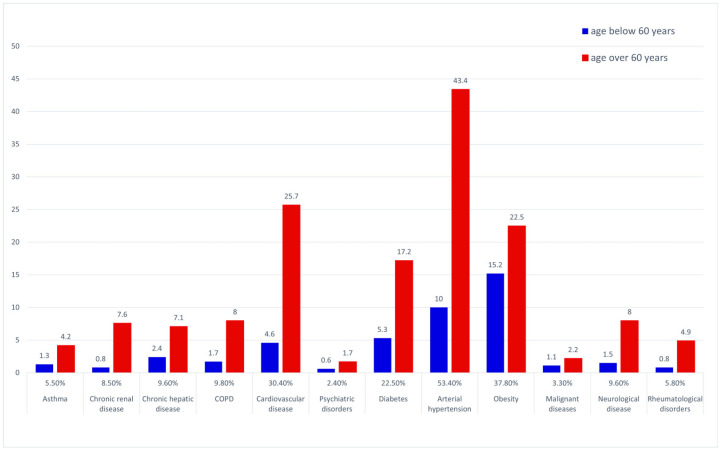
Prevalence of underlying co-morbid conditions—general data pool and age-stratified subgroups.

**Table 1 jpm-13-00900-t001:** Clinical features of patients hospitalized with moderate COVID-19 (Alpha and Delta outbreaks).

Clinical Feature	Present at Admission
cough	312 (69%)
shortness of breath	250 (55.3%)
fatigue	219 (48.4%)
fever	203 (44.9%)
myalgia	96 (21.2%)
headache	95 (21%)
anorexia	73 (16.1%)
chest pain	62 (13.7%)
diarrhea	51 (11.2%)
smell loss	51 (11.2%)
nausea	50 (11%)
arthralgia	45 (9.9%)
sore throat	41 (9%)
abdominal pain	24 (5.3%)
taste loss	21 (4.6%)
confusion	20 (4.4%)
nasal discharge	15 (3.3%)
convulsions	3 (0.6%)
wheezing	3 (0.6%)
skin rash	0 (0%)

**Table 2 jpm-13-00900-t002:** Outcomes of moderate-severity COVID-19 cases hospitalized in the presence of confusion and shortness of breath.

		Recovered	Transferred	Deceased	*p* Value (Chi Square)
confusion	absent	350 (77.7%)	41 (9.1%)	39 (8.6%)	<0.001
	present	8 (1.7%)	6 (1.3%)	6 (1.3%)	
shortness of breath	absent	176 (39%)	6 (1.3%)	19 (4.2%)	<0.001
	present	183 (40.6%)	41 (9.1%)	26 (5.7%)	

**Table 3 jpm-13-00900-t003:** Differences between age-stratified subgroups of moderate-severity COVID-19 patients in terms of clinical feature.

	Age ≤ 60 Years	Age > 60 Years	Age ≤ 60 Years (%)	Age > 60 Years (%)	*p* Value (Chi Square)
cough	119	190	67.2	70.1	
shortness of breath	76	174	42.9	64.2	<0.001 *
fatigue	72	146	40.6	54.4	0.005
fever	90	111	50.8	40.9	0.025
myalgia	41	53	23.1	19.7	
headache	42	52	23.7	19.3	
anorexia	24	48	13.6	17.8	
chest pain	22	39	12.4	14.4	
diarrhea	17	34	9.6	12.5	
nausea	19	31	10.7	11.5	
arthralgia	17	28	9.6	10.3	
smell loss	30	20	17	7.5	0.002
confusion	4	16	2.2	5.9	
abdominal pain	9	14	5	5.1	
sore throat	27	13	15.2	4.8	<0.001 *
taste loss	13	8	7.7	3.1	0.028
runny nose	7	8	3.9	2.9	
wheezing	1	2	0.5	0.7	
convulsions	2	1	1.1	0.3	

*—significant after Bonferroni correction.

**Table 4 jpm-13-00900-t004:** Logistic regression model including age group, the presence of confusion and shortness of breath at admission and outcome for moderate-severity COVID-19 cases requiring hospitalization.

	Odds Ratio	95% IC	*p* Value
confusion	5.73	2.17–15.17	0.0001
shortness of breath	2.08	1.22–3.53	0.007
age > 60 years	3.29	1.81–5.97	0.0001

**Table 5 jpm-13-00900-t005:** Significant differences in pre-existing comorbid conditions in the presence of confusion and shortness of breath at admission for moderate-severity COVID-19 cases requiring hospitalization.

		Shortness of Breath Absent	Present	*p* Value (Chi Square)
Arterial hypertension	absent	117 (26.1%)	91 (20.3%)	<0.001 *
	present	83 (18.5%)	157 (35%)	
Cardiovascular disease	absent	156 (34.8%)	156 (34.8%)	0.001 *
	present	45 (10%)	93 (20.7%)	
COPD	absent	189 (42.1%)	217 (48.4%)	0.016
	present	12 (2.6%)	32 (7.1%)	
		Confusion absent	present	*p* value (chi square)
Neurological disease	absent	391 (87.2%)	11 (2.4%)	<0.001 *
	present	35 (7.8)	8 (1.7%)	
Hepatic disease	absent	391 (87.2%)	14 (3.1%)	0.001 *
	present	37 (8.2%)	6 (1.3%)	
Cardiovascular disease	absent	302 (67.4%)	9 (2%)	0.024
	present	127 (28.3%)	11 (2.4%)	
Arterial hypertension	absent	203 (45.3%)	4 (0.8%)	0.02
	present	224 (50%)	16 (3.5%)	

*—significant after Bonferroni correction.

## Data Availability

Additional data are available on request from the corresponding author. More information about the unCover project can be obtained by accessing the official site uncover-eu.net.

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
