# Peer review of "Prognostic Role of Clinical Features of Moderate Forms of COVID-19 Requiring Hospitalization"

_jpm, 2023, doi:10.3390/jpm13060900_

Round 1

Reviewer 1 Report

The short manuscript describes and presents clinical features related to COVID prognosis in two regional Romanian respiratory disease centers. Although straightforward in approach, it requires significant revisions. Changes include but not limited to the following:

1. At least one Figure is encouraged.

2. Underlying health conditions such as chronic lung disease, cancer, diabetes, and obesity have a critical role in COVID response. Authors in the manuscript should include or describe the related information.

3. The p values in the manuscript were adjusted for multiple testing?

Average.

Author Response

Dear reviewer,

Thank you for the time and effort that went into reviewing our manuscript. Changes were made following your suggestions. Please find enclosed point-to-point responses to the issues that have been raised.

  1. At least one Figure is encouraged.

Two figures were added – relevant to point 2.

  1. Underlying health conditions such as chronic lung disease, cancer, diabetes, and obesity have a critical role in COVID response. Authors in the manuscript should include or describe the related information.

Usable information was limited to the database available under the unCover project which contained limited data on co-morbidities – large disease groups and no staging details (particularly important for hypertension, heart failure or COPD). Available data was added, and text was amended.

  1. The p values in the manuscript were adjusted for multiple testing?

Bonferroni was used and mentioned in the table legend; differences were considered significant only if they passed correction; if not they were reported as possibly significant. Text was amended for clarity.

Reviewer 2 Report

Trofor et al, in the current manuscript, analyzed symptoms and signs of moderate cases of COVID-19 that necessitate hospitalization, with the potential to identify factors that can predict unfavorable outcomes. Here authors analyzed data from alpha and delta variants outbreaks. Authors identified that worse outcomes were significantly linked to the presence of confusion, shortness of breath, and being over 60 years old. Trofor et al conclude that the clinical presentation upon admission could potentially serve as a prognostic factor for moderate cases of COVID-19. However, I have the following reservations regarding the conclusion,-

1) Line 30: Please follow the correct nomenclature- it is SARS-CoV-2

2) The authors have considered very limited confounding factors while analyzing the data. Have they considered co-morbidities while analyzing the symptoms post-SARS-CoV-2 infection?

3) Shortness of breath could result from poor cardiovascular health, smoking, COPD, anemia, or a body mass index over 30 (overweight). A respiratory infection could worsen this particular symptom. Therefore, it is important to know how many patients had underlying dyspnea.

4) Have authors considered patients who were repeatedly infected with SARS-CoV-2?

5) Can authors comment on Long COVID and the association of symptoms?

Authors could benefit from copy-editing. 

Author Response

Dear reviewer,

Thank you for the time and effort that went into reviewing our manuscript. Changes were made following your suggestions. Please find enclosed point-to-point responses to the issues that have been raised.

1) Line 30: Please follow the correct nomenclature- it is SARS-CoV-2

Thank you for this important reminder – the text was amended.

2) The authors have considered very limited confounding factors while analyzing the data. Have they considered co-morbidities while analyzing the symptoms post-SARS-CoV-2 infection?

Usable information was limited to the database available under the unCover project which contained limited data on co-morbidities – large disease groups and no staging details (particularly important for hypertension, heart failure or COPD). Available data was added, and text was amended.

3) Shortness of breath could result from poor cardiovascular health, smoking, COPD, anemia, or a body mass index over 30 (overweight). A respiratory infection could worsen this particular symptom. Therefore, it is important to know how many patients had underlying dyspnea.

Available data was added to the results section. Due to the constraints of retrospective nature of our dataset it is not possible to clearly determine the clinical status of the patients prior to admission.

4) Have authors considered patients who were repeatedly infected with SARS-CoV-2?

Although it would have been interesting to have such comparators. no such cases are available in the database. All data originates from different patient records – from the first viral episode (to the best of our knowledge).

5) Can authors comment on Long COVID and the association of symptoms?

Unfortunately no, locally available data is not relevant to long COVID.

Round 2

Reviewer 1 Report

Accept in present form.

Reviewer 2 Report

The authors have sufficiently addressed the questions raised.